# Knowledge, attitude, and practice of Bangladeshi residents during COVID-19 pandemic

**Mili Saha**[1], **Goutam Saha**[2]*, **Mynul Islam**[3]

**1** Department of English, Jagannath University, Dhaka, Bangladesh, **2** Department of Mathematics, University of Dhaka, Dhaka, Bangladesh, **3** Institute of Statistical Research and Training, University of Dhaka, Dhaka, Bangladesh

* ranamath06@gmail.com

**Data Availability Statement:** All data are presented in the manuscripts.

**Funding:** The authors received no specific funding for this work.

## Abstract

Bangladeshi government has adopted some special steps to control the quick spread of the COVID-19 pandemic situation. However, the residents' knowledge, attitudes, and practices towards the disease directly impact the success of the controlling measures taken by the state. This article explores knowledge ($K$) about preventions, attitude ($A$) to the disease, and practices ($P$) of preventing the COVID-19 infection risks of different age groups residing in Bangladesh. Quantitative data were collected online using a KAP questionnaire from 932 participants. Also, statistical $t$ and $F$ tests have been used and analyzed and $p$-value, 95% Confidence Interval, Odd Ratio (OR), KAP scores, and multiple logistic regression analysis, are presented in this research. Results show the population is generally aware of the symptoms and social distancing. They are concerned about re-spreading and positive about staying home. The most significant findings of the study reveal that the old age group (age 50 or over) is the most alert group, male population are the most vulnerable with less care, people living outside Dhaka take less care and fewer preventive measures against the deadly virus, the young age group (age 18–25) is most optimistic while the female respondent group is best prepared among all the participants.

## 1. Introduction

Bangladesh detected her first COVID-19 infected patients on 8 March 2020, which began with three in number. However, it has increased to 2456 on day forty-one, which was the sixth week and indicates the fourth phase of infection [1] called 'Sustained Human-to-Human Transmission' [2]. The total number of deaths has reached 100 including ten deaths and around 300 new infections every day, which indicates the rapid rise period. The government has locked down the highly affected areas and a general leave has been declared to keep all the private and public schools, businesses, and service farms closed. Despite all these steps taken by the state, COVID-19 is still a rising infectious health issue and should be prevented perceptively by individuals. This is more about personal preclusion than authority care to reduce the infection rate. Knowledge, attitudes, and practices towards infectious diseases usually involve some

**Competing interests:** The authors have declared that no competing interests exist.

panic emotions among the population, promoting complications in preventing the disease and spread [3]. Hence, understanding the public awareness level and readiness to combat the outbreak of COVID-19 in Bangladesh is crucial during the rapid rise period.

KAP refers to knowledge (what is known), attitude (what is thought), and practices (what is done). A KAP survey of any community about a particular topic serves as an educational diagnosis of that population through examining what people know, what they believe, and how they behave [4–6]. KAP is a very common tool used in health-seeking research [7]. This KAP study explores what Bangladeshi residents know about COVID-19 symptoms and prevention, how they view the socio-cultural effects of the disease, and what practices they use to prevent the infection. Understanding the KAP of the residents might enable the government authorities and other agencies to introduce and implement "*a more efficient process of awareness creation programs*" which will appropriately and effectively address the interventional needs of the community [8].

We performed this study to better understand the residents' perceptions and beliefs, which might identify any knowledge gaps, negative attitudes, and factors influencing actions. No studies report about this community's KAP of COVID-19 and the information essential to decide about the protective steps and select the program priorities. This research could be significantly useful for the local administration to reintegrate more impactful policies and successful measures to control infections and improve safety during and after the lock-down period. Bangladesh is almost at the peak of infection and the people have been continuing to fight against Coronavirus disease. To ensure that they win, "*people's adherence to the control measures are essential, which is largely affected by their knowledge, attitudes, and practices (KAP)*" [9, 10].

People are, overall, willing to maintain social distance and quarantine which can slow down the infection, although they have high anxiety about possible infection, which can be reduced through increasing awareness and addressing mental health issues [11]. Most of the participants are young at the age of the Covid-19 studies and 'younger individuals are more likely to be asymptomatic when infected and could be unaware they are putting others at risk' [12]. On the other hand, the chances of infection and the severity of illness are much direr with aged people [13]. Hence, hasty lifting of lock-down can promote a secondary peak while lifting lock-down gradually can flatten it [14]. Studying individual awareness can itself motivate people to practice the preventive measures discussed in the research alerting them in turn and know about the practices of a large population to avoid mass contamination.

Inadequate understanding of the common people about the nature and effects of a viral disease can result in improper practices and less adherence to hygiene rules, delayed treatment, rapid spread, and fatal consequences of the infection. Hence, improving the knowledge, awareness, and perceptions of all population is crucial [15]. Studies analyzing the current level of awareness, attitudes, and practices towards COVID 19 among Nepali, Saudi Arabian, Ethiopian, Pakistani, Nigerian, and Indian [16–22] residents emphasize ensuring people's willingness and active participation to minimize the pandemic effects. Bangladeshi male population, young generations, and rural participants have less knowledge, pessimistic attitudes, and malpractices towards the disease. Whereas, the female, elderly, and urban population have better knowledge and positive practices [23]. Some recent works on COVID-19 using the KAP model are also performed by many researchers [24–26].

## 2. Research design and methodology

The current research was conducted in Bangladesh and an online survey was performed using one of the popular Google tools called Google Forms. The link to the questionnaire was saved

for future use. We provided ethical clearance and ensured the participants that the given personal information and opinion will be kept confidential. The research questions include three aspects of the pandemic spreading rapidly in the country:

a. What do the Bangladeshi residents know about the symptoms and cure?

b. How do they view the social impacts and spreading possibilities of the virus?

c. What actions do the respondents perform to practice prevention?

## Participants

Most of these participants have a minimum educational background which is a Higher Secondary School Certificate level and use any one of the online platforms, such as Facebook, WhatsApp, Gmail. Our goal was to reach a bigger audience including a group of participants to elicit maximum responses.

## Ethics statement

Our survey study is performed following the authentic research guidance and protocol of the University of Dhaka, Dhaka-1000, Bangladesh and Jagannath University, Dhaka-1100, Bangladesh. This research survey is approved by the Chairman, Department of Mathematics, University of Dhaka, who gave us verbal consent to continue the present research. We also assured each of the participants that all the personal or institutional information gathered in this study will be used for academic publication purposes and their identities will never be exposed as it is stated in the ethical protocol. The consent was received from the participants through Google form and only those who agreed participated in this research survey.

## Tools

The survey questionnaire consists of two sections, including demographics and KAP inquiries. At the beginning, social demographic variables, such as age, gender, and place of residence (Dhaka vs. other districts in Bangladesh) were sought. Following the clinical and community management of COVID-19 guidelines by the Institute of Epidemiology, Disease Control and Research (IEDCR), Bangladesh, we prepared a COVID-19 awareness questionnaire including 14 questions (**S1 Questionnaire**). Five of the questions are about clinical arrangements (K1-K5), three of them are about the possible spreading medium (A1-A3), and six of them involve prevention and control (P1-P6) of the virus. The questions are answered on a three-item Likert scale consisting of yes, no, not sure options. We assigned 3 points for the first option, 2 points for the second option, and 1 for every third. The total knowledge score ranged from 1 to 30, which denotes better knowledge with higher scores. The Cronbach's alpha coefficient was 0.79 indicating acceptable internal consistency of the KAP questionnaire.

## Data collection

The primary data on the participants' attitudes for the online cross-sectional study were collected from 17 April 2020 to 20 April 2020. In this research, 565 male and 367 female in a total of 932 participants responded to the online survey. Among all, 574 participants belong to the age group ranging from 18 to 25 years, 349 of them belong to the age group ranging from 26 to 49 years, and only 9 participants are in the age group of 50 years and above. Also, 679 participants identify themselves as single and the remaining 253 of them are married. Besides, 514

respondents live in the Dhaka division, the most infected region in the country, and 418 participants live outside the Dhaka division.

Details of the above-mentioned various data types are also presented in the following:

| Demographic Information's (N = 932) | | |
|---|---|---|
| | Number | % |
| Gender: | | |
| Man | 565 | 60.62 |
| Woman | 367 | 39.38 |
| Age group: | | |
| 18–25 | 574 | 61.58 |
| 26–49 | 349 | 37.45 |
| 50 and 50+ | 9 | 0.97 |
| Marital Status: | | |
| Single | 679 | 72.85 |
| Married | 253 | 27.15 |
| Place of Residence: | | |
| Dhaka division | 514 | 55.15 |
| Outside Dhaka Division | 418 | 45.85 |

For calculating sample size, the proportion of the population having adequate knowledge about COVID-19 was considered as the indicator variable. The expected sample size was calculated as at least 576 where the z score for 95% confidence interval was 1.9, the prevalence of adequate knowledge was assumed 0.5, the margin of error was 0.05 and the design effect was 1.5 for sampling variation. We provided the participants with the link using different online platforms including Facebook, Messenger, Google talk, email, and requested them to share the link with their friends and relatives on any social media so they can respond as well. In addition, we requested the participants' demographic information comprising gender identity, marital status, and place of residence. Therefore, participants were included in the study through social network communities using snowball sampling. All the data were available in the following link: https://docs.google.com/spreadsheets/d/1Qndd80zdSltt_9W1x9uKsieUCASreqyYh8yFEfpjYkk/edit?usp=sharing.

## Data analysis

Later the data have been examined and categorized by the percentage of the agreement, disagreement, and personal preferences in each question. Also, a statistical analysis using statistical software SPSS with a 5% significance level was done and a two-sided T-test are considered for statistical analysis of the data. In addition, statistical F test is also used and $p$-value, 95% Confidence Interval (CI), Odd Ratio (OR), KAP scores, and multiple logistic regression analysis are also presented.

## 3. Results and discussion

### 3.1 KAP scores

The participants' knowledge, attitude, and practice score are shown using descriptive statistics in Table 1. The KAP scores are converted to a quantitative scale and samples are independent across the demographic variables. An independent sample t-test and ANOVA have been applied to understand whether the average participants' knowledge, attitude, and practice

**Table 1. Descriptive statistics of knowledge, attitude, and practice scores.**

| Variable | N | Minimum | Maximum | Mean | Standard Deviation |
|---|---|---|---|---|---|
| Knowledge score | 932 | 7 | 15 | 11.15 | 1.41 |
| Attitude score | | 3 | 9 | 6.43 | 1.35 |
| Practice score | | 7 | 17 | 12.41 | 1.75 |

towards COVID 19 scores significantly vary across gender, location, and age group. The normality assumption for implementing ANOVA has been checked through a q-q plot for each group. Thus, the comparison of knowledge, attitude, and practice score for demographic variables such as gender, location, and age group are shown in Table 2. The average knowledge score found for the participants was 11.15 (Table 1). The average knowledge scores of the male participants are higher than the female participants average. However, this difference is not statistically significant (p-value > 0.05). The average attitude towards the COVID-19 situation score was 6.43 (Table 1). Participants' attitude scores range between 3 to 9 with SD 1.34. The analysis also demonstrates that the average attitude score does not vary significantly across the gender, location, and age group of the participants. This result also coincides with the results obtained in Maheshwari et al. [20]. The average practice score was 6.43 (Table 1). Participants' practice scores range between 7 to 17 with SD 1.75. Participants' practice scores vary significantly across gender, location, and age group of the participants. Male participants have a greater tendency to practice the preventions compared to the female ones for the safety against possible COVID-19 infections. And, the differences among the practice scores of the male and female participants are statistically significant at a 1% level. The Covid-19 prevention practicing tendency is higher among elder age groups. Besides, the difference in practice scores among different age groups is statistically important at a 1% level of significance. Observation shows Dhaka divisional participants have slightly higher practice scores than the participants residing outside this division with considerable differences at a 5% level of significance. The majority of the participants (n = 652, 69.7%) have adequate knowledge. Following the independent sample t-test and ANOVA, it can be stated that the average knowledge score does not vary considerably across the location and age group of the participants at a 5% level of significance. Similar kinds of results are found by Maheshwari et al. [20].

### 3.2 Distribution of KAP

**Knowledge. K1:** In response to the first survey question, 78.76% of respondents report complete awareness about the varied symptoms of COVID-19 infection and common flu while 14.80% are still unsure. And, 6.44% are not aware of the differences at all, which can be either fatal or stressful for them. The findings reveal that female respondents and old age groups (50 years and above), and the population living in greater Dhaka are more aware of the differences between the symptoms of COVID-19 and other common flues than their counterparts who are most confused about the COVID-19 symptoms too.

**K2:** Also, the majority of people (59.76%) are unsure about any proposed or available medicines that can be effective to treat the infected persons. And, 24.36% of them do not rely on any medicine for treating the disease while 15.88% believe if they are infected, these medicines can help. This confirms the other results demonstrating a particular number of people perceive the severity through media focus, which might be wrong in reality. More male respondents, young age groups (18–25), and people residing outside Dhaka believe in the medicines presented in various media than the female, elder, and Dhaka resident participants while the most hesitant population is the old aged group.

**Table 2. Comparison of knowledge, attitude, and practice scores among different demographic variables.**

| Variables | Knowledge Score | | |
| --- | --- | --- | --- |
| | Mean±SD | t/F value | p-value |
| Gender | | | |
| Woman | 11.05±1.36 | -1.62 | 0.11 |
| Man | 11.21±1.43 | | |
| Location | | | |
| Dhaka division | 11.17±1.36 | 0.58 | 0.56 |
| Outside Dhaka Division | 11.11±1.46 | | |
| Age group (Years) | | | |
| 18–25 | 11.06±1.44 | 2.66 | 0.07 |
| 26–49 | 11.26±1.36 | | |
| 50+ | 11.56±0.88 | | |
| | Attitude Score | | |
| Gender | | | |
| Woman | 6.36±1.32 | -1.26 | 0.21 |
| Man | 6.48±1.36 | | |
| Location | | | |
| Dhaka division | 6.41±1.35 | -0.475 | 0.64 |
| Outside Dhaka Division | 6.45±1.34 | | |
| Age group (Years) | | | |
| 18–25 | 6.42±1.33 | 0.57 | 0.57 |
| 26–49 | 6.46±1.38 | | |
| 50+ | 6.00±1.22 | | |
| | Practice Score | | |
| Gender | | | |
| Woman | 12.11±1.58 | -4.173 | <0.001*** |
| Man | 12.60±1.83 | | |
| Location | | | |
| Dhaka division | 12.52±1.69 | 2.07 | 0.04* |
| Outside Dhaka Division | 12.28±1.82 | | |
| Age group (Years) | | | |
| 18–25 | 12.22±1.74 | 8.12 | <0.001*** |
| 26–49 | 12.70±1.73 | | |
| 50+ | 12.55±1.23 | | |

*p < 0.05

**p < 0.01

***p < 0.001.

**K3:** A great majority of the respondents (77.68%) maintain social distancing by being 1.5 meters away from other people while 8.91% of them keep less distance and the rest 13.41% are unsure about the practice, which defines the overall careless movement of the community. Again, the female, elderly, and non-Dhaka residents groups are found to maintain social distance more than their male, young, and Dhaka-residents counterparts who are most confused about maintaining the required social distance to avoid infection.

**K4:** Findings suggest that a considerable major group of 64.48% of respondents believe getting closer to the infected persons is the worst potential source of COVID-19 transmission in Bangladesh while secondary contact through infected person's used items is the next possible

**Table 3. Distribution of knowledge related characteristics among the participants.**

| Type | Respondent Types | A (%) | B (%) | C (%) |
|------|------------------|-------|-------|-------|
| | Gender | | | |
| K1 | Male | 75.22 | 9.03 | 15.75 |
| K2 | | 19.47 | 24.25 | 56.28 |
| K3 | | 74.87 | 10.27 | 14.87 |
| K4 | | 4.25 | 66.90 | 28.85 |
| K5 | | 42.22 | 54.89 | 2.89 |
| K1 | Female | 84.2 | 2.45 | 13.35 |
| K2 | | 10.35 | 24.52 | 65.12 |
| K3 | | 82.02 | 6.81 | 11.17 |
| K4 | | 7.36 | 60.76 | 31.88 |
| K5 | | 43.10 | 54.88 | 2.02 |
| | Age groups | | | |
| K1 | 18–25 | 76.48 | 14.14 | 9.38 |
| K2 | | 16.55 | 22.82 | 60.63 |
| K3 | | 77.53 | 8.19 | 14.29 |
| K4 | | 5.75 | 65.51 | 28.75 |
| K5 | | 57.97 | 39.34 | 2.69 |
| K1 | 26–49 | 81.95 | 14.05 | 4.0 |
| K2 | | 15.19 | 27.22 | 57.59 |
| K3 | | 77.65 | 10.03 | 12.32 |
| K4 | | 5.16 | 62.75 | 32.09 |
| K5 | | 47.88 | 49.42 | 2.70 |
| K1 | 50 and over | 100 | 0 | 0 |
| K2 | | 0 | 11.11 | 88.89 |
| K3 | | 88.89 | 11.11 | 0 |
| K4 | | 0 | 66.67 | 33.33 |
| K5 | | 66.67 | 33.33 | 0.0 |
| | Place of residence | | | |
| K1 | Dhaka division | 82.30 | 5.05 | 12.65 |
| K2 | | 13.82 | 24.90 | 61.28 |
| K3 | | 78.99 | 8.56 | 12.45 |
| K4 | | 6.23 | 63.62 | 30.16 |
| K5 | | 43.98 | 52.58 | 3.44 |
| K1 | Outside Dhaka division | 74.40 | 8.13 | 17.46 |
| K2 | | 18.42 | 23.68 | 57.90 |
| K3 | | 76.08 | 9.33 | 14.59 |
| K4 | | 4.55 | 65.55 | 29.90 |
| K5 | | 40.76 | 57.48 | 1.76 |

where A, B, and C indicate three possible options for each knowledge question.

risk of contamination. And, a very small group of 5.48% of participants are concerned with air transmission, which indicates this group's extreme awareness about the virus's existence in the environment. Also, it is found that 18–25 years and 26–49 years old age groups have comparable beliefs about the local sources of contamination, although the female participants tend to believe more in air transmission and infection through touching things used by infected

persons than the male counterparts. The old age group completely ignores the possibility of air transmission.

**K5:** A smaller majority of the respondents (42.51%) prefer to take all preventive measures than testing (54.81%) if they discover COVID-19 symptoms in them. The worst risk is still 2.67% of respondents plan to hide the infection news from others, which can be detrimental for their surroundings. Both male and female groups are equally interested in trying preventive measures, testing, and hiding about infections. The old age group is most interested in preventive measures; non-Dhaka residents are mostly aware of testing; Dhaka-residents are prone to hide infections most, although the old age group is not at all interested in hiding.

A summary of the percentage results for the knowledge section is presented in Table 3.

**Attitude. A1:** The research participants are generally anxious about the negative social attitudes, negligence, and rudeness showed towards the COVID-19 patients in Bangladesh. Data shows 48.06% of the respondents are frustrated, 40.24% of them are afraid and 11.70% are angry to see such insolence and humiliation including abundance, refusal, and rejection to treatment, and avoidance by family and society. The groups have similar reactions to the negative social attitudes, rudeness, and negligence shown to the Corona patients in Bangladesh, although the old age group is more angry and frustrated, who are least scared with this attitude.

**A2:** A great majority of the respondents (49.25%) are found to be concerned that the virus will re-spread as soon as the lock-down is relaxed. A very minor group (13.84%) of the respondents believe the situation will become normal while 36.91% of them are unsure about the choices. The female participants are more anxious about COVID-19 re-spreading after the lock-down is lifted and are less optimistic about normalizing the situation than the male respondents. Also, the old age group is least concerned about re-spreading, least hopeful about normalizing, and most confused about choosing any of the relevant options.

**A3:** Frequently disinfecting things and washing hands are the second priority to the participants (29.08%) while keeping a social distance is the highest priority to the 47.42% of respondents, which the World Health Organization emphasizes. Although wearing safety masks and gloves is the least preferred measure after the lock-down period, this is the most common practice at this rapid rise period. However, male participants outperform the female respondent group regarding the post-lock-down awareness. Data shows the male participants group is more likely to wear safety masks and gloves in addition to maintaining social distance while the female respondents are prone to washing hands and disinfecting things more frequently than their male counterparts. Also, young participants groups and non-Dhaka residents outperform the counter old aged and Dhaka-resident groups with more awareness about the upcoming risks.

A summary of the percentage results for the attitude section is presented in Table 4.

**Practice. P1:** Despite being remarkably alert about keeping social distance and avoid close contacts with the community members, the majority of the research population (81.65%) are relaxed about eating animal protein including eggs, meat, or fish. Whereas, 7.84% of participants are seriously aware about such transmission through animal bodies and they have been avoiding eggs and milk too. Also, another group consisting of 10.51% respondents avoid eating meats and fish. Therefore, the majority of people are at great risk in case the local animal transmission begins. Surprisingly old age groups are least concerned about consuming animal proteins to prevent COVID-19 infection. Next comes the female participant group who avoid less animal protein than the male population. And, Dhaka residents avoid eating animal protein more than the residents living outside Dhaka. Gender and ages significantly influence the participants' practices about protecting themselves from COVID-19 infection and treatment of the disease.

**Table 4. Distribution of attitude related characteristics among the participants.**

| Type | Respondent Types | A (%) | B (%) | C (%) |
|---|---|---|---|---|
| | Gender | | | |
| A1 | Male | 10.97 | 48.85 | 40.18 |
| A2 | | 48.67 | 15.22 | 36.11 |
| A3 | | 26.73 | 49.38 | 23.89 |
| A1 | Female | 12.81 | 46.87 | 40.33 |
| A2 | | 50.14 | 11.72 | 38.15 |
| A3 | | 22.89 | 44.41 | 32.70 |
| | Age groups | | | |
| A1 | 18–25 | 11.15 | 46.34 | 42.51 |
| A2 | | 46.69 | 15.33 | 37.98 |
| A3 | | 23.69 | 50.70 | 25.61 |
| A1 | 26–49 | 12.32 | 50.43 | 37.25 |
| A2 | | 53.87 | 11.75 | 34.38 |
| A3 | | 23.21 | 42.12 | 34.67 |
| A1 | 50 and over | 22.22 | 66.67 | 11.11 |
| A2 | | 33.33 | 0.0 | 66.67 |
| A3 | | 22.22 | 44.44 | 33.33 |
| | Place of residence | | | |
| A1 | Dhaka division | 13.04 | 48.05 | 38.91 |
| A2 | | 50.39 | 14.20 | 35.41 |
| A3 | | 23.35 | 43.19 | 33.46 |
| A1 | Outside Dhaka division | 10.05 | 48.09 | 41.87 |
| A2 | | 47.85 | 13.40 | 38.76 |
| A3 | | 23.68 | 52.63 | 23.68 |

where A, B, and C indicate three possible options for each attitude question.

**P2:** Since 52.36% of participants prefer to stay home and 34.23% of the respondents go out once a week, it seems those 13.41% of participants who go for daily shopping are at a greater risk. As usual, male respondents, mid-young age (26–49 years) group, and non-Dhaka resident group go out of the home more frequently than the female, early-young, Dhaka-resident, and old age groups. However, female and old age groups mostly stay home.

**P3:** Another positive preventative measure that a great majority of the population (67.17%) avails is always disinfecting things and taking a bath right after returning home. However, 27.04% of respondents take this measure less frequently and 5.79% of the population do it seldom, which puts the risk of contaminating the home environment unintentionally and reduces the chance of escaping infection. Female, mid-young age, and Dhaka-residents groups are more careful about taking bath and disinfecting things after coming back from the outside than the counter male, early-young, non-Dhaka residents, and old groups, while the male group is the least and old age group is the most aware regarding the precaution.

**P4:** The majority of the participants (51.06%) only avoid cold food items and drinks as prevention from the infection, although 44.56% of them are more cautious about drinking warm water to stop the virus living inside the body and 4.38% only take steam as they think it can help them to prevent the virus. It is found that more female, mid-young aged, and non-Dhaka resident participants avoid cold food and drinks. A greater number of male, old-aged and Dhaka-resident participants drink warm water. More male, old-aged and non-Dhaka resident respondents take steam to prevent COVID-19 infection than their counter groups. Although

female groups are quite aware of avoiding cold foods and drinks, they are the least responsive group to drinking warm water and taking steam.

**P5:** The critical aspect of this research findings include that a considerable number of participants (43.13%) rely on home care and nearly an equal 41.63% of those who prefer to shift to hospital initially after being detected as COVID-19 positive. This indicated the local populations' reliance on the state policies and physicians' availability for treating COVID-19 patients. However, 15.24% of the population is indecisive regarding the matter, making it chaos and delaying the cure of the infected individuals. The mid young age groups are most prone to stay home when they are infected while early young age groups prefer shifting to the hospital for Covid-19 treatment. And, old age groups are the most indecisive which is risky too. Also, Dhaka residents rely more on home care and the people residing outside Dhaka prefer to move to the hospital than relying on homecare. However, gender shows no considerable correlation to choosing COVID-19 treatment facilities among the respondents.

**P6:** Regarding social services, such as raising social awareness and community care, 77.86% of participants mainly use the online platform and social media to ensure social distancing while 15.16% try to make people aware through practicing the health rules, and 6.98% report about demonstrating the safety tasks to instruct people how to save from infecting. Responses regarding raising mass awareness during the pandemic period demonstrate the groups' almost equal share of work using any particular mode of demonstration.

A summary of the percentage results for the practice section is presented in Table 5.

The unusual situations and unexpected circumstances created by the coronavirus pandemic have undoubtedly changed the livelihoods, attitudes, and priorities at the local and global levels. Bangladeshi residents are no exceptions. Since the coronavirus outbreak, people have been loaded with massive information about different aspects of the virus infections, affects, and cure during the last few months. Some might be exaggerated and unnecessary, while some other essential information might be missing. These include symptoms, precautions, preventions, infection types, treatment, recovery, risks, deaths, etc. Although a vast majority of the local and global population are well informed about the essential aspects, the high-risk population including youth groups, males, and marginalized communities need to be educated and aware [21]. Hence, more awareness and preparation programs should be introduced regarding COVID-19 for the people residing in remote and underdeveloped areas where mass communication through online technology is limited. People need to continue to strengthen KAP towards COVID-19 to win the current and future battles against the disease. Policymakers need to put more emphasis on informing and awakening the less educated, low-income, and male young [17]. Besides, the government needs to design educational sessions for less knowledgeable people to enhance their knowledge [19].

Generally, as the findings show, educated people are supportive of the adopted measures to combat and control COVID-19 spread. Although, low-income people have no way out to maintain all such measures. Almost 80% of citizens understand and enforce the necessity of undergoing complete isolation, disinfection, and prevention. However, people including day laborers and front liners might expect the authorities to provide protective masks and other staff for all citizens, and local governments have scopes to consider such an initiative. As the current studies expose, 10–20% population is a bit careless and aware of the symptoms, social distancing, outing, and protections in the different parts of the world, especially the young man and peripheral groups have some resistances to extreme caution. Although the possibility of individual contraction has not been reduced, some citizens have a low level of concern. Many adult men do not properly maintain cleaning and disinfecting but move around the need to be aware of maintaining protection. They adopt far fewer safeguard measures practically than women, so these populations should communicate further regarding COVID-19

**Table 5. Distribution of practice related characteristics among the participants.**

| Type | Respondent Types | A (%) | B (%) | C (%) |
|------|------------------|-------|-------|-------|
| | | Gender | | |
| P1 | Male | 10.44 | 11.33 | 78.23 |
| P2 | | 19.82 | 43.54 | 36.64 |
| P3 | | 57.70 | 34.51 | 7.79 |
| P4 | | 49 | 45.7 | 5.3 |
| P5 | | 42.65 | 41.59 | 15.75 |
| P6 | | 14.39 | 7.19 | 78.42 |
| P1 | Female | 3.81 | 9.26 | 86.92 |
| P2 | | 3.54 | 19.89 | 76.57 |
| P3 | | 81.74 | 15.53 | 2.72 |
| P4 | | 54.15 | 42.86 | 2.99 |
| P5 | | 43.87 | 41.69 | 14.44 |
| P6 | | 16.34 | 6.65 | 77.0 |
| | | Age groups | | |
| P1 | 18–25 | 8.36 | 10.80 | 80.84 |
| P2 | | 11.50 | 31.01 | 57.49 |
| P3 | | 64.11 | 29.44 | 6.45 |
| P4 | | 50.93 | 44.93 | 4.14 |
| P5 | | 31.71 | 51.39 | 16.90 |
| P6 | | 13.36 | 6.15 | 80.49 |
| P1 | 26–49 | 7.16 | 10.32 | 82.52 |
| P2 | | 16.62 | 39.83 | 43.55 |
| P3 | | 72.49 | 22.64 | 4.87 |
| P4 | | 52.45 | 43.02 | 4.53 |
| P5 | | 62.18 | 26.07 | 11.75 |
| P6 | | 17.94 | 8.53 | 73.53 |
| P1 | 50 and over | 0 | 0 | 100 |
| P2 | | 11.11 | 22.22 | 66.67 |
| P3 | | 55.56 | 44.44 | 0 |
| P4 | | 0 | 83.33 | 16.67 |
| P5 | | 33.33 | 22.22 | 44.44 |
| P6 | | 25.0 | 0.0 | 75.0 |
| | | Place of residence | | |
| P1 | Dhaka division | 8.37 | 10.51 | 81.13 |
| P2 | | 9.34 | 35.21 | 55.45 |
| P3 | | 74.51 | 20.82 | 4.67 |
| P4 | | 48.29 | 47.80 | 3.9 |
| P5 | | 46.50 | 37.35 | 16.15 |
| P6 | | 15.08 | 6.35 | 78.57 |
| P1 | Outside Dhaka division | 7.18 | 10.52 | 82.30 |
| P2 | | 18.42 | 33.01 | 48.57 |
| P3 | | 58.13 | 34.69 | 7.18 |
| P4 | | 54.36 | 40.70 | 4.94 |
| P5 | | 39.0 | 46.89 | 14.11 |
| P6 | | 15.25 | 7.75 | 77.0 |

where A, B, and C indicate three possible options for each practice question.

risk management. Young adults need effective health education campaigns enhancing and encouraging knowledge, a positive mindset, and essential preventions of COVID-19 even more [27].

Overall, age is significantly correlated to inadequate knowledge, inappropriate practices, and poor perceptions about COVID-19 in the United Arab Emirates [15]. Especially, male and youth groups need to be empowered through acknowledging their preventive practices against COVID-19 and the sense of responsibilities in Bangladesh [23]. However, the current findings contradict Hayat's [18] conclusion that increasing age is associated with poor knowledge and hence, poor practices. Also, gender has noticeable effects on positive attitudes and good practices, such as avoiding crowded places in Saudi Arabia, Pakistan, and India, and the Philippines [17, 19–21]. Although female participants are also more aware regarding many aspects, including cleaning, disinfecting, and further outbreak, they are less active in self-care and rarely avoid animal meat or fish which might be unsafe. Rural residency, low educational status, and poor income significantly correlate to insufficient knowledge and bad practices in Pakistan and India [19, 20]. The youth age group is more optimistic while the female respondent group is prepared most of all participants.

### 3.3 Regression analysis

Multiple logistic regressions are performed and shown in Table 6 to find out the adjusted effect of the potential risk factors on adequate knowledge, positive attitude, and good practice on COVID-19. To perform logistic regression, mean KAP score is considered as cutoff score to define sufficient knowledge, positive attitudes, and good practices. In multiple logistic regressions, the covariates are taken on the basis of having a significant association or relation with the outcome variables. According to the independent sample t-test, significant relations between gender, location, and age group with knowledge, attitude, and practice score have been found. Thus, these variables are included in logistic regression analysis. The odds of having adequate knowledge do not vary significantly across the gender and location of the participants, as shown in Table 2. Moreover, Table 2 shows the participants of the mid-age group 26–49 are more likely (1.41 times) of possessing adequate knowledge than the participants of the young age group 18–25. The odds of having adequate knowledge for the participants of age 50

**Table 6. Estimated parameters of multiple logistic regression on different factors.**

| Variables | Sufficient Knowledge | | Positive Attitudes | | Good Practices | |
|---|---|---|---|---|---|---|
| | n (%) | OR adjusted (95% CI) | n (%) | OR adjusted (95% CI) | n (%) | OR adjusted (95% CI) |
| Gender | | | | | | |
| Woman | 250 (68.1) | *Ref.* | 269 (73.3) | *Ref.* | 242 (65.9) | *Ref.* |
| Man | 420 (71.2) | 1.19 (0.89–1.59) | 421 (74.5) | 1.05 (0.78–1.42) | 404 (71.5) | 1.43* (1.06–1.90) |
| Age-group | | | | | | |
| 18–25 | 385 (67.1) | *Ref.* | 428 (74.6) | *Ref.* | 377 (65.7) | *Ref.* |
| 26–49 | 259 (74.2) | 1.41* (1.04–1.90) | 255 (73.1) | 0.93 (0.69–1.27) | 261 (74.8) | 1.50** (1.11–2.02) |
| 50+ | 8 (88.9) | 3.74 (0.46–30.25) | 7 (77.8) | 1.22 (0.25–5.98) | 8 (88.9) | 3.48 (0.43–28.19) |
| Location | | | | | | |
| Outside Dhaka | 286 (68.4) | *Ref.* | 313 (74.9) | *Ref.* | 268 (64.1) | *Ref.* |
| Dhaka | 366 (71.2) | 1.11 (0.84–1.49) | 377 (73.3) | 0.94 (0.69–1.26) | 378 (73.5) | 1.56** (1.17–2.08) |

*p < 0.05

**p < 0.01

***p < 0.001.

or above is 3 times higher than the participants of age group 18–25, although this difference is not statistically significant. A high percentage of the participants (n = 690, 73.7%) showed positive attitudes towards COVID-19 disease. Multiple logistic regressions reveal that there is no association between positive attitudes and participant characteristics variables such as gender, age group, and location (Table 6). A large proportion of the participants (n = 646, 69%) had good practice towards COVID-19. A significant difference between male and female participants in terms of good practice at a 5% significance level. Male participants are 43% more likely to practice prevention than the female participant's group. Participants with age group 26–49 have a 1.50 times higher chance of performing good practices towards COVID-19 than the participants with age group 18–25. The odds of doing good practices for the participants of age 50 or above is 3.48 times higher than the participants of age group 18–25, although this difference is not statistically significant. Participants in the Dhaka division are more intended to do good practice towards COVID-19 than that of the participants outside the Dhaka division. Therefore, it is recommended that women, young age groups, and people residing outside Dhaka should pay more attention to practice prevention against COVID-19.

## 4. Conclusion

Findings show Bangladeshi women and old age groups are generally socially and psychologically connected, although physically distanced. And thus, they are preventing the risks of infection. However, young age groups are comparatively less aware of the symptoms, social distancing, and frequent outings, which indicates the greater risks involved with the male population and those living in divisions other than Dhaka. The most significant findings of the study reveal that the old age group is the most alert group, male respondents are the most vulnerable with less care. People living outside Dhaka have less knowledge and fewer preventive measures against the deadly virus. The young age group is more optimistic while the woman respondent group is prepared for most of all participants. Hence, more awareness and preparation programs should be introduced regarding COVID-19 for the people residing in remote and peripheral areas where mass communication through online technology is limited. The overall findings demonstrate that 16–20% of respondents comprising mainly the young man groups are at high risk of infection. Also, the analysis of the survey results confirms a correlation between the knowledge and practice of COVID-19 protection in Bangladesh.

### Recommendations

Recommendations are presented in the following:

1. Communication and information strategies should be adapted and updated through different communication channels, including print and television media, to reach the at-risk age and gender groups. In addition to television and print media, authorities can utilize digital media to reach the young age groups in rural, peripheral, and urban areas for adapting communication strategies since they are more active online.

2. A considerable number of populations in every country are unaware of the most recognizable symptoms and preventions of COVID-19, potentially creating new clusters and increasing the rate of infections. Hence, this is undeniably advisable to inform this population including youth groups, males, patients, and peripheral residents about the possible contamination. Confidence should be raised among the female and the elderly groups who are more preventive and aware than the male and young groups.

3. Also, targeted health education should be introduced as a response strategy to COVID-19 and contextually relevant in low-income settings. Enhancing KAPs among extremely poor

communities is crucial and challenging as tailored guidance for public health response for the LMICs.

4. COVID-19 knowledge may include many irrelevant issues, rumors, and general misconceptions affecting the short- and long-term control efforts against the disease. Hence, these should be incorporated in targeted training and campaigns.

5. Besides, COVID-19 information "*coming from informal sources such as close persons outweighs information that comes from other sources. Propaganda is important, but word of mouth is what is more trusted*" [28]. Notably, young and highly educated people like to check information sources. The government should use credible sources, including registered phones and emails, to communicate with citizens who are often skeptical about the information received through OTT communication platforms, such as Viber, WhatsApp, etc. so the communication strategy should be adapted taking this fact into account. Some NGOs, namely UNDP, UNICEF, or WHO, are generally more trusted sources of information than traditional or modern media. The government and media can cooperate and collaborate with these profiles.

6. In addition, possible job loss or global recession has raised emotions, such as uncertainty, fear, and concern which should be considered in designing necessary communication strategies. Besides, communicating objective news regarding the pandemic scale, setting a priority of COVID-19 testing, and the ways to recover the Corona affected finances and job-seekers career can minimize the mass doubts and increase trust about the real situations and adopting the measures.

7. Finally, mobilizing KAPs towards COVID-19, especially for the chronic disease sufferers including cardio and asthma patients coming from remote, low income, old age, and low educational backgrounds. Leaflets containing detailed information about COVID-19 and written in local languages should be administered among the patients and their families.

## Limitation of the study

Some limitations of this research are presented below:

1. Data were collected within a high-rise period of covid-19. The time period can be extended to get more responses.

2. We were not able to reach the research survey to a high number of the population due to lockdown.

## Supporting information

**S1 Questionnaire.**
(DOCX)

## Author Contributions

**Conceptualization:** Goutam Saha.

**Data curation:** Goutam Saha, Mynul Islam.

**Formal analysis:** Mili Saha.

**Investigation:** Mili Saha.

**Methodology:** Goutam Saha.

**Software:** Goutam Saha, Mynul Islam.

**Supervision:** Goutam Saha.

**Visualization:** Goutam Saha.

**Writing – original draft:** Mili Saha, Goutam Saha.

**Writing – review & editing:** Mili Saha, Mynul Islam.

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
