## [Decision Letter · Decision Letter 0]

29 Nov 2021

PGPH-D-21-00907

Knowledge, Attitude, and Practice of Bangladeshi Residents during COVID-19 Pandemic

Dear Dr. Saha,

Thank you for submitting your manuscript to PLOS Global Public Health. After careful consideration, we feel that it has merit but does not fully meet PLOS Global Public Health’s publication criteria as it currently stands. Therefore, we invite you to submit a revised version of the manuscript that addresses the points raised during the review process.

We look forward to receiving your revised manuscript.

Kind regards,

Carl Abelardo T. Antonio

Academic Editor

Journal Requirements:

1. PLOS ONE does not copy edit accepted manuscripts (https://journals.plos.org/globalpublichealth/s/criteria-for-publication#loc-5). To that effect, please ensure that your submission is free of typos and grammatical errors.

*Please change "female” or "male" to "woman” or "man" as appropriate, when used as a noun (see for instance https://apastyle.apa.org/style-grammar-guidelines/bias-free-language/gender).

2.  If you have no competing interests to declare, please state "The authors have declared that no competing interests exist"

Additional Editor Comments (if provided):

I concur with your peers that further description and explication of the methods and results is required for this manuscript before it is even considered for publication. While the changes required are extensive, I do think that these can be completed within the turnaround time of the journal.

Reviewers' comments:

Reviewer's Responses to Questions

**Comments to the Author**

1. Does this manuscript meet PLOS Global Public Health’s publication criteria? Is the manuscript technically sound, and do the data support the conclusions? The manuscript must describe methodologically and ethically rigorous research with conclusions that are appropriately drawn based on the data presented.

Reviewer #1: Partly

Reviewer #2: Partly

2. Has the statistical analysis been performed appropriately and rigorously?

Reviewer #1: No

Reviewer #2: I don't know

3. Have the authors made all data underlying the findings in their manuscript fully available (please refer to the Data Availability Statement at the start of the manuscript PDF file)?

Reviewer #1: Yes

Reviewer #2: No

4. Is the manuscript presented in an intelligible fashion and written in standard English?

Reviewer #1: No

Reviewer #2: No

5. Review Comments to the Author

Reviewer #1: The following are my comments concerning the manuscript:

- It should be checked by an editor for grammatical errors to improve its readability

- In the abstract, the authors specified that they used logistic regression and correlation analysis in their study. However, the main text did not mention this in the data analysis section.

- The authors should explain the rationale for employing statistical tests in their study. In particular, what objectives necessitate the use of the tests they presented.

- If logistic regression is necessary, specify the variable selection procedure they used to determine the final model

- Authors should mention whether the assumptions of the statistical tests were met and how were these tested.

- Study design was not explicitly specified.

- Sample size computation was not described.

- The section concerning data collection included some information that should be placed under the Results section.

- Tables must be reformatted to include only important statistics. For instance, t- and F-values are unnecessary since p-value and confidence interval estimates were already reported.

- There are too many tables. Consider deleting or merging some of the tables.

- Consider adding sub-sections to the results section to streamline its contents.

- Too many abbreviations were used. I suggest that the KAP questions be explicitly stated in the tables instead of using abbreviations.

Reviewer #2: 1. The ethics review and approval process was not clearly and adequately described in the manuscript.

2. Please expound in the methodology section what statistical tests were used for which specific objective or outcome of the study.

3. It is recommended that the results presented in Table 8 be discussed in the first part of the results section.

4. What are the cut-off values for the variables knowledge, attitudes and practices? (i.e. What is the cutoff score for knowledge to be considered as good?)

5. The tabular presentation of data needs to be revised because the way that it is presented is not clear (e.g. what do the column headers A, B, C mean?). Finally, the labels for the tables need to be revised and include more elements (e.g.variable, statistic, population, time period, sample size).

6. What does the phrase in the results and discussion and conclusion sections "the old age group is the most alert group" mean?

7.The discussion section did not introduce the salient findings of the study. Furthermore, the results of the study were not adequately discussed in the section. What do the results imply? How is it compared to other studies? Comparison of the current study's findings were briefly addressed not in the discussion section, but in the results section. What are the limitations of the study? Also, the recommendations section can still be reduced.

8. In the discussion section, studies conducted in Pakistan and India were referenced but no in-text citations were included.

9.The article needs proofreading to improve its readibility. There are some parts of the manuscript were words seem to be randomly inserted (e.g. the word "inquires" in the first sentence under the research design subsection 'tools'; the words "p value" and "counter" in A3)

10. The in-text citation and reference list do not follow the journal's recommended format (Vancouver)

6. PLOS authors have the option to publish the peer review history of their article (what does this mean?). If published, this will include your full peer review and any attached files.

**Do you want your identity to be public for this peer review?** For information about this choice, including consent withdrawal, please see our Privacy Policy.

Reviewer #1: **Yes: **Kim L. Cochon

Reviewer #2: No

---

## [Editor Report · Decision Letter 1]

6 Apr 2022

Knowledge, Attitude, and Practice of Bangladeshi Residents during COVID-19 Pandemic

PGPH-D-21-00907R1

Dear Dr Saha,

We are pleased to inform you that your manuscript 'Knowledge, Attitude, and Practice of Bangladeshi Residents during COVID-19 Pandemic' has been provisionally accepted for publication in PLOS Global Public Health.

Best regards,

Carl Abelardo T. Antonio

Academic Editor

Thank you for taking time to address the reviewers' comments.